# Nanometric Cu-ZnO Particles Supported on N-Doped Graphitic Carbon as Catalysts for the Selective CO_2_ Hydrogenation to Methanol

**DOI:** 10.3390/nano14050476

**Published:** 2024-03-06

**Authors:** Lu Peng, Bogdan Jurca, Alberto Garcia-Baldovi, Liang Tian, German Sastre, Ana Primo, Vasile Parvulescu, Amarajothi Dhakshinamoorthy, Hermenegildo Garcia

**Affiliations:** 1Instituto de Tecnología Química, Consejo Superior de Investigaciones Científicas-Universitat Politecnica de Valencia, Av. De los Naranjos s/n, 46022 Valencia, Spain; lu.peng@mpikg.mpg.de (L.P.); baldovi.alber@gmail.com (A.G.-B.); ltian@itq.upv.es (L.T.); gsastre@itq.upv.es (G.S.);; 2Department of Organic Chemistry, Biochemistry and Catalysis, University of Bucharest, B-dul Regina Elisabeta 4-12, 030016 Bucharest, Romania; bjurca@gw-chimie.math.unibuc.ro; 3Departamento de Química, Universitat Politècnica de València, C/Camino de Vera, s/n, 46022 Valencia, Spain; admguru@gmail.com

**Keywords:** heterogeneous catalysis, CO_2_ hydrogenation, N-doped graphene, methanol synthesis

## Abstract

The quest for efficient catalysts based on abundant elements that can promote the selective CO_2_ hydrogenation to green methanol still continues. Most of the reported catalysts are based on Cu/ZnO supported in inorganic oxides, with not much progress with respect to the benchmark Cu/ZnO/Al_2_O_3_ catalyst. The use of carbon supports for Cu/ZnO particles is much less explored in spite of the favorable strong metal support interaction that these doped carbons can establish. This manuscript reports the preparation of a series of Cu-ZnO@(N)C samples consisting of Cu/ZnO particles embedded within a N-doped graphitic carbon with a wide range of Cu/Zn atomic ratio. The preparation procedure relies on the transformation of chitosan, a biomass waste, into N-doped graphitic carbon by pyrolysis, which establishes a strong interaction with Cu nanoparticles (NPs) formed simultaneously by Cu^2+^ salt reduction during the graphitization. Zn^2+^ ions are subsequently added to the Cu–graphene material by impregnation. All the Cu/ZnO@(N)C samples promote methanol formation in the CO_2_ hydrogenation at temperatures from 200 to 300 °C, with the temperature increasing CO_2_ conversion and decreasing methanol selectivity. The best performing Cu-ZnO@(N)C sample achieves at 300 °C a CO_2_ conversion of 23% and a methanol selectivity of 21% that is among the highest reported, particularly for a carbon-based support. DFT calculations indicate the role of pyridinic N doping atoms stabilizing the Cu/ZnO NPs and supporting the formate pathway as the most likely reaction mechanism.

## 1. Introduction

Methanol is among the most valuable products that can derive hydrogenation from CO_2_ [1,2,3]. Being in the liquid state at ambient conditions, methanol has other important advantages compared to alternative products formed in CO_2_ hydrogenation, including water solubility, non-corrosiveness, high volumetric energy content, and easy transformation into gasoline [4,5] and aromatics [6,7], among other chemicals [8,9]. Although methanol is currently produced on a large multi-ton scale, and there was an estimated 100 millions of metric tons produced in 2020, there is still the possibility to considerably increase methanol production [10], particularly if application of methanol as a fuel or hydrogen carrier is finally implemented [11,12,13]. In any case, market forecasts indicate that methanol production will at least double by 2030 [12].

Since formic acid has a low H_2_ content and is corrosive, another important advantage of methanol vs. formic acid is its much higher mass and volumetric energy content, about four times higher than that of formic acid [14,15]. Methanol can be used directly as transportation fuel in combustion engines [15] and it can also be directly used as liquid fuel in proton exchange membrane fuel cells [16]. Methanol is also considered a liquid organic hydrogen carrier [17,18], with a H_2_ storage capacity of about 19 wt.% [18,19]. Although methanol reforming will still emit unwanted CO_2_, the cycle could have a zero CO_2_ footprint if methanol is in turn formed from CO_2_ [20]. Equation (1) corresponds to methanol synthesis from CO_2_ hydrogenation.
(1)$${CO}_{2}+3H_{2}\to {CH}_{3}OH+H_{2}O\;\;\;\;\Delta H=-49.5kJ/mol$$

As indicated in Equation (1), the partial CO_2_ hydrogenation to methanol is an exothermic reaction, with the equilibrium towards methanol formation being more favorable at low temperatures and high pressures [21,22]. However, the slow reaction kinetics determine that in order to achieve measurable reaction rates, heating of the system and the use of suitable catalysts are required to form methanol.

At high temperatures, CO_2_ conversion can be limited by equilibrium composition. In addition, besides methanol, CO appears generally as a competing product (Equation (2)). Typical CO_2_ hydrogenation mixtures are composed of methanol and CO in various proportions accompanied by lesser amounts of methane, thus decreasing methanol selectivity. Formation of CO and CH_4_ prevails in CO_2_ hydrogenation at high temperatures.
(2)$${CO}_{2}+H_{2}\to CO+H_{2}O\;\;\;\;\Delta H=+41.2kJ/mol$$

Due to these constraints, CO_2_ hydrogenation to methanol is carried out at temperatures in a range from 250 to 300 °C and high pressures, for which a compromise between thermodynamics and kinetics requirements can be reached. Also, to overcome this thermodynamic limitation, photocatalytic CO_2_ reduction to methanol is gaining importance, including the use of g-C_3_N_4_ [22,23,24].

After the discovery by BASF of copper chromite (Cu-CrO_3_) as a catalyst for CO_2_ hydrogenation to methanol [25], the most widely used catalyst is copper and zinc oxide supported on alumina (Cu-ZnO/Al_2_O_3_), which is considered as the current benchmark catalyst for the process [25]. The typical atomic Cu/Zn proportion is 2:1 and loading on Al_2_O_3_ can be over 30 wt.% [26,27,28,29,30]. Although the catalyst is generally denoted as Cu-ZnO, in situ studies suggest that Cu-ZnO is a precursor of the active species formed under the reaction conditions by Cu restructuration, ZnO chemical reduction to Zn metal, and Cu-Zn alloying [31,32]. Thus, even though under ambient conditions Zn is present as oxide and Cu and Zn are in different phases, the as-prepared Cu-ZnO/Al_2_O_3_ should be considered a precursor of the active sites that have been proposed to be Cu NPs decorated by Zn atoms [32]. Besides alumina, zirconia in different crystallographic phases is considered also as a suitable support [33,34,35].

In spite of the fact that Cu-ZnO/Al_2_O_3_ was reported many years ago and considering the current intense research on catalytic CO_2_ hydrogenation to methanol [23,33,36], progress in the development of alternative, more efficient, catalysts that could promote methanol synthesis from CO_2_ at lower temperatures is still unsatisfactory [33] or based on less abundant elements [37,38,39]. Among the non-containing-Cu catalysts, indium oxide can promote CO_2_ hydrogenation to methanol with a high selectivity [37,38,39]. Besides facets of In_2_O_3_, oxygen vacancies are the active sites [40]. Pd doping increases the activity of In_2_O_3_ without much negative influence on the selectivity [41]. However, indium is considered as a scarce element, particularly compared with abundant Cu and Zn, and it is included in the list of critical raw materials to be avoided [42]. 

In a series of articles, we have been showing that N-doped graphitic carbons are suitable supports to develop highly selective Fe-Co catalysts for various CO_2_ hydrogenation reactions, including the Sabatier reaction [43], the reverse water gas shift (RWGS) (Equation (2)) [44], and for the formation of C_2+_ products [45]. In this context, it is also of interest to expand the use of N-doped graphitic carbon matrices as supports for Cu-ZnO NPs [Cu-ZnO@(N)C] and to determine their catalytic activity for methanol synthesis under operation conditions compatible with the thermodynamic limitations of the process. In this way, the materials here prepared based on chitosan derived from biomass wastes represent a clear example of waste valorization, applying circular economy principles. The transformation of chitosan into a N-doped graphitic carbon, supporting metal NPs to be used as catalysts, considerably increases the value of the biomass waste. 

In comparison to inorganic oxides, carbonaceous supports for Cu-ZnO have been significantly less studied, with most of these studies being limited to carbon nanotubes for methanol synthesis from CO/H_2_ [46] and steam reforming of methanol [47]. In one of the few precedents on the use of carbon supports, Cu-CuO and ZnO were formed in a porous carbon FDU-15 obtained by pyrolysis of resol, observing a similar performance for CO_2_ hydrogenation to methanol than the benchmark BASF catalyst [48]. Therefore, the catalytic activity of Cu-ZnO@(N)C samples still appears to be worth exploring. As it will be shown below, the experimental data support that Cu-ZnO@(N)C is an efficient, selective, and stable catalyst for partial CO_2_ hydrogenation to methanol, resulting in a notable methanol productivity of 83 g_CH3OH_ kg_catalyst_^−1^ h^−1^.

## 2. Materials and Methods

### 2.1. Synthesis of Samples Cu@(N)C and Cu-ZnO@(N)C

Cu@(N)C (sample **1**) and Cu-ZnO@(N)C (samples **2**–**6**) were obtained by dissolving 1 g chitosan with 625 μL acetic acid in 50 mL Milli-Q water. After chitosan dissolved completely, the solution was introduced dropwise, with a syringe (0.8 mm diameter needle), in an aqueous solution of sodium hydroxide (500 mL, 2 M). The hydrogel microspheres were formed immediately and immersed in NaOH solution for 2 h and then profusely washed with distilled water to pH 7. Afterwards, the resulting hydrogel microspheres were washed by a series of ethanol/water baths with an increasing concentration of ethanol (10, 30, 50, 70, 90, 100 vol.%, respectively) for 15 min in each and immersed in 100 mL Cu(OAc)_2_–ethanol solution with different concentrations, as indicated in Appendix A, for 2 days with a slow stirring, then washed with anhydrous ethanol, and subsequently dried by supercritical CO_2_. Drying using supercritical CO_2_ ensures high porosity and large surface area of the aerogel microspheres in comparison with alternative drying procedures [44]. The resulting aerogel microspheres were pyrolyzed under Ar flow (200 mL/min), increasing the temperature at a rate of 2 °C/min up to 200 °C for 2 h and then to 900 °C for 2 h. The as-prepared samples did not exhibit pyrophoric properties. The resulting Cu@(N)C was ground into powder and immersed in 30 mL Zn(OAc)_2_–ethanol solution with different concentrations for 2 days with slow stirring. After removal of ethanol at 60 °C overnight, the Zn^2+^-containing Cu@(N)C was heated at a rate of 2 °C/min up to 200 °C for 2 h to obtain the final Cu-ZnO@(N)C.

### 2.2. Preparation of Cu-ZnO/Al_2_O_3_

Copper and zinc oxide supported on alumina (Cu-Zn/Al_2_O_3_) were prepared by impregnation method in two or one steps. Al_2_O_3_ powder (270 mg) was dispersed into an ethanol solution (20 mL) of Cu(OAc)_2_ (254.5 mg) and the suspension was stirred at room temperature overnight until the solvent evaporated. Then, the obtained powder was pyrolyzed under Ar flow (200 mL/min), increasing the temperature at a rate of 2 °C/min up to 200 °C for 2 h and then to 900 °C for 2 h. After cooling at room temperature, the resulting Cu/Al_2_O_3_ powder was impregnated in ethanol solution (20 mL) with Zn(OAc)_2_ (10 mg) and the suspension was stirred at room temperature overnight until the solvent evaporated. The resulting powder was annealed at a rate of 2 °C/min up to 200 °C for 2 h under Ar flow (200 mL/min). Another analogous Cu-Zn/Al_2_O_3_ sample in which Cu-impregnated Al_2_O_3_ was not submitted to pyrolysis was also prepared (Cu-Zn/Al_2_O_3_-wp, wp meaning without pyrolysis). A third sample was prepared as Cu-Zn/Al_2_O_3-_wp except that the impregnation of Cu(OAc)_2_ and Zn(OAc)_2_ was carried out with the same amounts and times but in a single step (Cu-Zn/Al_2_O_3_-imp).

### 2.3. Sample Characterization

X-ray diffraction (XRD) patterns were obtained in a Philips XPert diffractometer (Košice, Slovakia) (40 kV and 45 mA) equipped with a graphite monochromator employing Ni-filtered Cu Kα radiation (1.541178 Å). Raman spectra were collected with a Horiba Jobin Yvon-Labram HR UV-visible–NIR (Kyoto, Japan) (200–1600 nm). Raman microscope spectrometer model had a 514 nm laser. The carbon and nitrogen content of the samples was determined by combustion chemical analysis by using a CHNS FISONS elemental analyzer (Spain). The chemical analysis was determined by ICP-OES (iCAP 7400, Thermo Scientific, Waltham, MA, USA) from the mother liquor after digesting the Cu-ZnO@(N)C samples in aqua regia at 60 °C for one day. High-resolution field emission scanning electron microscopy (HR-FESEM) images were acquired by using a Zeiss GeminiSEM500 apparatus (Jena, Germany). High-resolution transmission electron microscope (HR-TEM) images were recorded in a JEOL JEM 2100F (Košice, Slovakia) under an accelerating voltage of 200 kV coupled with an X-Max 80 energy-dispersive X-ray detector (Oxford instruments, Abingdon, UK). This HR-TEM is equipped with dark-field and high-angle field image detectors that facilitate the observation of phase contrast with different atomic numbers. Samples for measurement were prepared by dropping a few drops of the suspended material in ethanol or dichloromethane on a carbon-coated nickel grid and drying at room temperature overnight. The average metal particle size was determined by measuring the diameter of a statistically relevant number of metal NPs in dark field TEM images using the program J-image. The results of this measurement are presented in the furthest-right column of Table 1.

### 2.4. Computational Models and Methods

Periodic DFT calculations were conducted using the Cambridge Serial Total Energy Package (CASTEP) module with the exchange–correlation functional described by Perdew–Burke–Ernzerhof (revised version for solids) within the generalized gradient approximation (GGA-PBEsol). Tkatchenko and Scheffler (TS) dispersion corrections, scheme, were incorporated along with the exchange and correlation functional to improve the structural and vibrational properties. Slabs were separated by 15 Å along [001], perpendicular to the surface. A self-consistent field method (tolerance 1.0 × 10^−7^ eV/atom) was employed in conjunction with plane-wave basis sets with a cutoff energy of 500 eV in reciprocal space. All structures were geometry optimized until energy was converged to 1.0 × 10^−6^ eV/atom, maximum force to 0.025 eV/Å, and maximum displacement to 5.0 × 10^−3^ Å.

The transition states of every elementary reaction in the CO_2_ hydrogenation process were obtained by the complete linear synchronous (LST) and quadratic synchronous transit (QST) methods. The adsorption energy of species over the ZnCu/N-C models was calculated as E_b_ = E_total_ − E_A_ − E_sur,_ where E_total_ represents the total energy of the catalytic surface with the adsorbed molecule and E_A_ and E_sur_ are the energies of isolated adsorbate molecule and the clean surface, respectively. The energy of an isolated molecule (E_A_) is computed by placing it in the same lattice box (about 15 × 15 × 18 Å^3^). The activation energy barriers (E_a_) of every step of the elementary reactions are defined as E_a_ = E_TS_ − E_IS_, where E_IS_ and E_TS_ are the total energy of the initial state (IS) and transition state (TS), respectively.

### 2.5. Catalyst Testing

Samples **1**–**6** were tested for CO_2_ hydrogenation in a PID Eng&Tech setup (Microactivity Spain) equipped with a stainless steel (316 SS) fixed-bed tube microreactor (Autoclave Engineers) featured with an inner K-type thermocouple in contact with the catalyst packed bed. Flow rates of reagent gases (H_2_ 5.0-Linde and CO_2_ 4.5-Linde) were controlled with two calibrated mass flow controllers (EL-FLOW Select-Bronkhorst, Nijverheidsstraat, The Netherlands). At least two independent catalytic tests were performed for each sample.

For each test, 40 mg of catalyst powder was introduced in the reactor, air was removed by flushing the system at room temperature for 15 min with ten times the flow rates used during the catalytic tests, followed by 10 min at the flow rates used during the experiments (3.2 mL·min^−1^ H_2_, 0.8 mL·min^−1^ CO_2_). The reactor was afterwards slowly pressurized at 40 bars without changing the flow rates of the gas reagents. Two successive GC injections were performed with the gases passing through the reactor at room temperature to check the stability of the gas-phase composition inside the pressurized reactor. Four reaction temperatures from 150 to 300 °C with steps of 50 °C were investigated. For each temperature, GC injection was performed 90 min after the stabilization of the temperature to achieve a steady-state regime of the reactor setup.

Gas samples from the reactor output were passed through a transfer line kept thermostated at 110 °C to be analyzed using an Agilent 7890A gas chromatography instrument with TCD detection. Product separation was performed using capillary PLOT columns: molecular sieve (RT-Molsieve 5A-Restek, for CO, CO_2_, and light hydrocarbons) and divinylbenzene (SupelQ-Supelco, for methanol) with H_2_ carrier gas (linear velocities between 30 and 41 cm/s). Each gas sample was injected through a remote-controlled 6-way valve (A4C6WE-Vici, thermostated at the same temperature as the transfer line), with a 90 μL injection loop. The system is operated under continuous flow and some of the experiments are ran for several tens of hours to ensure that the products formed do not derive from the catalyst.

## 3. Results

### 3.1. Sample Preparation and Characterization

Initial attempts to prepare Cu-Zn@(N)C were carried out following similar preparation procedures as those described in related precedents on Fe-Co@(N)C [43,44] consisting of the pyrolysis of chitosan powders embedding simultaneously Co and Fe metal ions. During the pyrolysis, chitosan, a polysaccharide of glucosamine, becomes converted in turbostratic N-doped graphitic carbon that can be completely exfoliated to single- or few-layer defective graphene upon sonication [49]. The strong reductive conditions of the hot carbon material during the pyrolysis under inert atmosphere are responsible for the reduction of metal ions to the metallic state [50]. In the present case, chitosan solutions were impregnated with a mixture of Cu(OAc)_2_ and Zn(OAc)_2_ in various molar ratios. However, due to the low boiling point of Zn metal and the flushing Ar flow, these attempts with Cu-Zn@(N)C were met with failure. Complete Zn evaporation occurred under the conditions of pyrolysis, with only Cu on the (N)C support remaining after pyrolysis.

To overcome this limitation, ZnO was incorporated after the pyrolysis of Cu^2+^ salts adsorbed on chitosan. While it would be possible also to simultaneously incorporate Cu and Zn after chitosan pyrolysis and formation of the N-doped graphitic carbon, it was anticipated that the strong interaction between the Cu metal NPs and the graphene sheets of the carbon resulting from the pyrolysis of Cu^2+^-chitosan [51] would be lost if the sample was prepared by impregnation of N-doped graphitic carbon by Cu(OAc)_2_. Data in the literature have shown that pyrolysis at 900 °C of Cu^2+^ ions adsorbed on chitosan renders a material in which the resulting Cu NPs become strongly grafted on the (N)C support, as deduced from the relatively small particle size, the flat morphology of the Cu NPs, their preferential (111) facet orientation matching the graphene structure, and binding energy shifts in XPS [51,52,53]. Therefore, in the present study, we proceeded to incorporate Cu(OAc)_2_ to the chitosan before pyrolysis to obtain Cu@(N)C and, subsequently, to impregnate the desired Zn(OAc)_2_ amount on the preformed Cu@(N)C. It was reasoned that in this way the interaction between the Cu NPs and defective graphene sheets of the carbon matrix as a substrate would be strong [51] and subsequent deposition of Zn(OAc)_2_ could still reconstruct the Cu-Zn alloy under the reaction conditions. Figure 1 illustrates the steps performed in the preparation of Cu-ZnO@(N)C. Further details can be found in the experimental section.

A series of samples with different atomic Cu/Zn ratios were prepared, trying to cover a wide range of Cu/Zn ratios, including a Cu/Zn ratio of around 3 that is close to the composition of the benchmark Cu-ZnO/Al_2_O_3_ catalyst. Appendix A indicates the exact weights of Cu(OAc)_2_ and Zn(OAc)_2_ used in the preparation of samples **1**–**6**. Unavoidably, the total Cu + Zn loading on the material varied from sample to sample due to poor control of the weight loss during pyrolysis, although for samples **3**–**6** the Cu + Zn loading was close to 10%. Note that sample **1** containing only Cu was the material to which Zn(OAc)_2_ was not added and, therefore, it has a lower total metal percentage. Table 1 summarizes the set of samples under study, the main Cu-Zn analytical data, and the average metal particle size. It is worth noting that although NaOH was used in the formation of Cu(OAc)_2_-impregnated chitosan beads, due to their water solubility and the sublimation of any possible residual Na during the pyrolysis, the Na content of the final Cu-ZnO@(N)G samples was negligible.

The percentages of Cu and Zn in the samples were determined by ICP-OES elemental analysis after treating the Cu@(N)C and Cu-ZnO@(N)C samples with *aqua regia*, quantifying the metal content of the digested liquor. These data summarized in Table 1 show that the Cu/Zn ratio ranges from ∞ for sample **1**, which does not contain ZnO, to 0.53 for sample **2**, which is the sample with the highest ZnO proportion. This variation in the Cu/Zn ratio allows one to gain information on the influence of this parameter on methanol selectivity. 

High-resolution TEM images show that the Cu@(N)C and Cu-ZnO@(N)C samples contain metal NPs deposited on 2D defective graphene sheets that constitute the graphitic carbon matrix. Figure 1 shows selected dark-field TEM images taken in three different areas for the samples under study, illustrating that the metal NPs are homogeneously distributed through the carbon matrix as a consequence of the preparation procedure. The white zones of the images indicate where the Cu and Zn metals are present, while the black background indicates the absence of these metals. These images show that the metals are spread out through the carbon matrix. The particle size distribution and the average dimension, ranging from 1.0 ± 0.2 to 1.4 ± 0.4 nm, were determined by measuring the size of a statistically relevant number of those metal NPs. Similar average particle size values for the series of samples are also collected in Table 1, while the corresponding particle size distribution histograms are inserted in the DF-TEM images presented in Figure 1. The absence of large NPs can be observed in the images by the absence of bright dots. This small dimension of the Cu-ZnO particles in spite of the relatively high loading of metal (up to 14 wt.%) reflects the occurrence of a strong interaction of the Cu NPs with the defective N-doped graphitic carbon that thwarts the growth of the Cu particle even though the pyrolysis is carried out at 900 °C. High-temperature annealing is known to cause agglomeration of small metal NPs, as has been observed in other cases [51]. Thus, the presence of N on the graphitic carbon matrix is a prerequisite to obtain small metal NPs, since in the absence of N doping the resulting particle size is considerably larger, even in the range of 100 nm [54]. There are precedents in the literature claiming the interaction of N atoms on graphene with supported metals, such as Pt resulting in the formation of single atoms or small clusters [55,56,57].

Samples **1**–**6** were also characterized by XRD and Raman spectroscopy (Appendix A). In the XRD patterns, the expected diffraction peaks corresponding to metallic Cu (PDF No. 70-3038) [58] and ZnO (JCPDS No. 36-1451) [59] could be clearly identified for samples **1**–**6**. Appendix A shows the XRD patterns, indicating the assignment of the peaks, either to Cu metal or ZnO. The relative intensity of the peaks corresponding to ZnO was in accordance with the relative proportion of ZnO in the sample with respect to Cu. In addition, Appendix A also shows the formation of graphitic carbon with a broad peak appearing around 25· in the XRD pattern. The Scherrer equation was used to quantify the size of Cu and ZnO NPs that are based on the full width at half height of the most intense peaks in XRD resulting in values of 7.8 and 25.0 nm, respectively. These average sizes are much larger than those determined by TEM, appearing in Table 1. This discrepancy could be due to the fact that XRD measures the most crystalline particles in the Cu-ZnO(N)C samples and the small metal particles observed in TEM do not contribute much to the XRD.

The defective nature of N-doped graphene was established by Raman spectroscopy, where the characteristic G and D bands appearing at 1590 and 1350 cm^−1^, together with resolved overtone 2D at 2700 cm^−1^, were recorded [60]. Appendix A plots the representative Raman spectra recorded for each sample **1**–**6**. The relative intensity ratio of the G vs. the D band was about 1.15 and their width at half height is in accordance with values for N-doped graphitic carbons previously reported from chitosan [60]. 

High-resolution FESEM images at the 100–400 nm scale recorded for the Cu@(N)C and Cu-ZnO@(N)C materials reveal a fluffy, poorly packed, and highly porous morphology of the graphitic carbon matrix acting as a support for the metal–metal oxide NPs. This porous structure is inherited from chitosan aerogel beads dried in super-critical CO_2_ in which aggregation of chitosan fibrils by hydrogen bridges has been minimized [61]. Figure 2 and Appendix A show selected HR-FESEM images for Cu@(N)C (sample **1**) and Cu-ZnO@(N)C samples **2**–**6**, illustrating the porous, coral-like structure of the carbonaceous matrix, resulting from the graphitization of the polysaccharide fibrils of the precursor. As expected, no metal NPs could be observed in the HR-FESEM images due to their lower resolution, in agreement with the nanometric particle size of Cu-ZnO NPs measured by HR-TEM.

To gain information on the interaction between the Cu-ZnO NPs and the (N)G support and to determine the oxidation state of the fresh Cu-ZnO@(N)C samples, XPS analysis of catalyst **4** was performed as a representative sample of the series. XPS analysis of sample **4** revealed the presence of the expected Cu, Zn, C, N, and O elements, but with remarkably different proportions on the surface compared to the analytical data of the bulk material. The elemental proportion based on XPS is provided in Appendix A, while the XPS peaks for the elements and their best deconvolutions are presented in Appendix A. As it can be seen there, the percentages of the Cu and Zn elements on the surface are much below the values expected by the bulk analysis, with C, O, and N of the graphitic carbon being the prevalent surface elements. This result is in accordance with the carbon matrix wrapping the metal NPs. In addition, the surface Cu/Zn XPS ratio is about 0.5, far from the 4.2 ratio measured for the bulk sample. Since samples **2**–**5** are obtained by Zn impregnation on preformed Cu@(N)C, it appears that the Zn element is more external than the originally introduced Cu metal due to it being incorporated in the samples last. Besides elemental composition, analysis of the high resolution XPS Zn 2p core level peak indicates that it corresponds well to a single ZnO component with a binding energy value of 1022 eV, in accordance with the literature [62]. In contrast, the Cu 2p spectra indicate two components attributable to Cu^0^ and Cu^II^ oxidation states appearing at 932.3 and 934.4 eV, respectively. These values are downshifted by 0.7 eV in the case of Cu^0^ and upshifted by 0.9 eV for Cu^II^, with respect to the reported literature values for these two oxidation states [63]. These shifts in the binding energy support the occurrence of a strong Cu-(N)G interaction due to the preparation procedure based on high-temperature graphitization, as has been reported earlier [64]. 

For the sake of comparison and to put into context the catalytic activity of samples **1**–**6**, three samples consisting of Cu-ZnO supported on Al_2_O_3_ (Cu-ZnO/Al_2_O_3_) were also prepared. A similar two-step impregnation procedure to that used for the preparation of Cu-ZnO@(N)C was followed for the preparation of Cu-ZnO/Al_2_O_3_, trying to reproduce the method employed in the preparation of Cu-ZnO@(N)C with Al_2_O_3_ as a support. A second Cu-ZnO/Al_2_O_3_-wp (wp meaning without pyrolysis) was also prepared, consecutively adsorbing Cu and Zn, but without submitting the sample to pyrolysis. A third Cu-ZnO/Al_2_O_3_ was prepared by simultaneous Cu and Zn salts impregnation and subsequent mild baking at 250 °C (Cu-ZnO/Al_2_O_3_-imp, imp meaning impregnation).

### 3.2. Catalytic Activity

The objective of the present study is to establish the performance of Cu-ZnO active sites supported on N-doped graphitic carbon as catalysts for the selective partial hydrogenation of CO_2_ to methanol, following the lead of previously reported Fe-Co@(N)C catalysts that exhibit very high selectivity for CO_2_ hydrogenation to methane [43], CO [44], or C_2+_ [45], depending on the metal particle size and composition. The use of inorganic supports has been widely studied in the literature [65,66], but there is a limited effort made to gain information about the performance of graphitic carbons as a support in metal catalysts for methanol formation from CO_2_ [67]. 

The catalytic experiments were carried out in a pressurized stainless-steel reactor operating at 40 bar under 4 mL of continuous flow and a H_2_ to CO_2_ ratio of 4. After considering the known thermodynamic limitations [24], the range of temperatures studied was between 150 and 300 °C in 50 °C steps that were maintained for 1 h before going to the next temperature increase. Systematic calculation of the Weisz–Prater number for each catalyst and conditions indicates that in none of the cases the reaction was under diffusion control (see Appendix A). Previous controls operating at 300 °C in the absence of a catalyst showed that the austenite stainless steel reactor converts 1.1% of CO_2_ with a selectivity to CO and CH_4_ of 92.4 and 7.6%, respectively. This low CO_2_ conversion and CO selectivity was maintained with the time of stream in a 3 h test and decreased with the reaction temperature. Appendix A lists the CO_2_ conversion and product selectivity of these previous control experiments in the absence of a catalyst at different temperatures. Methanol was undetectable in these control experiments.

Catalyst **1** (Cu@(N)C) containing only Cu did not promote methanol formation and catalyzed mostly the RWGS (Equation (2)), accompanied by some formation of CH_4_ and C_2+_ products, with a combined selectivity of about 18%. This relatively high proportion of hydrocarbons in catalyst **1** could indicate the occurrence of some Fischer–Tropsch synthesis in which CO undergoes further hydrogenation of these hydrocarbons. 

In contrast to the blank controls and the results with sample **1** lacking ZnO, methanol formation was observed in most of the reactions carried out in the presence of graphitic carbon-supported Cu-ZnO catalysts. The catalytic results achieved by sample **1** make clear that the presence of ZnO is required to drive the selectivity towards methanol. Analysis of the reaction products shows that besides the formation of methanol and CO as the major products, lesser amounts of CH_4_, C_2_H_6_, and C_3_H_8_ (below a combined percentage of 5%) are also formed (see Appendix A and Appendix A). It should be commented that although alkali metal ions can be promoters of the catalytic activity of transition metals in CO_2_ hydrogenation reactions and we have used NaOH during the preparation of the present Cu-ZnO@(N)G samples, the Na content in these samples is negligible.

To optimize the methanol formation, the performance of Cu-ZnO@(N)C samples was studied at different temperatures from 150 to 300 °C. Figure 3b shows the variation in CO_2_ conversion and product selectivity in the case of sample **4**, while the results of other samples are presented in Appendix A. As expected, CO_2_ conversion increased for all the catalysts upon increasing the temperature in the 150–300 °C range. Methanol selectivity showed the opposite trend, being higher at lower CO_2_ conversions and decreasing in favor of CO at higher temperatures. This general behavior agrees with the thermodynamics of the two main competing reactions presented in Equations (1) and (2), with methanol formation being exothermic and RWGS giving CO, being endothermic [24]. At temperatures of 350 °C or higher, CO_2_ conversions increase, but methanol selectivity becomes negligible or even methanol formation becomes undetectable. From the dependence of the CO_2_ conversion on the temperature (Appendix A), the apparent activation energies (E_a_) for CO_2_ hydrogenation in the range of temperatures between 150 and 300 °C were obtained for each catalyst of the series (see Appendix A). Except for sample **3**, the E_a_ was in the range of 50 to 43 kJ mol^−1^, with the lowest value being for sample **4**.

The series of catalysts using N-doped graphitic carbon as a support did not show a clear influence of the Cu/Zn ratio on methanol selectivity, which was mostly dependent on CO_2_ conversion. For the same CO_2_ conversion, similar methanol selectivity values were reached regardless of if the Cu/Zn ratio is high (sample **6**, Cu/Zn 8.1) or low (sample **2**, Cu/Zn 0.53). The main influence of the Cu/Zn ratio appears to be in CO_2_ conversion, which was the highest for sample **4**, the sample with the lowest E_a_. To illustrate differences in the catalytic performance depending on the Cu/Zn ratio, Figure 3a presents the CO_2_ conversion and selectivity for samples **1**–**6** working at 300 °C, 40 bar, at a H_2_/CO_2_ ratio of 4. As can be seen, besides differences in CO_2_ conversion, CO was the main product for all catalysts at 300 °C, although the formation of methanol was observed for all the series of Cu-ZnO@(N)C samples **2**–**6**. Appendix A provides CO_2_ conversion and selectivity for samples **1**–**6** at 200 and 250 °C. The relative activity order of samples **1**–**6** found at 300 °C is maintained at 200 and 250 °C, with sample **4** being the best performing catalyst. 

From the screening of the Cu-ZnO@(N)C catalysts under study shown in Figure 3a, sample **4** was the best performing catalyst, reaching at 200 °C a conversion of 1.7% with the maximum methanol selectivity of 89.7%. The highest methanol productivity achieved for sample **4** as a catalyst was 2.57 mol_CH3OH_ kg_catalyst_^−1^ h^−1^ or 82.24 g_CH3OH_ kg_catalyst_^−1^ h^−1^. To put these data into context, Appendix A provides a comparison of reported data in the literature. Although a comparison of catalyst performance measured under different conditions must be taken always cautiously, Appendix A shows that the methanol productivity achieved in the present study is comparable or overcomes those previously reported for the best catalysts for methanol synthesis from CO_2_. 

In fact, one key issue that is not reflected in Appendix A is catalyst stability. In the present case, sample **4** exhibits a remarkable catalytic stability, particularly in comparison to the benchmark Cu-ZnO/Al_2_O_3_. In a long run of 56 h under the reaction conditions, the catalytic activity of sample **4** did not decrease (Appendix A). Furthermore, after this long test at 300 °C, the 56 h used sample **4** exhibit, for two subsequent cycles of increased temperature from 150 to 300 °C in 1 h step of 50 °C temperature increase, identical catalytic data to those measured for the fresh sample presented in Appendix A. These data confirm again the catalyst’s stability. This catalyst stability for long reaction runs was also observed for catalysts 2 and 3. It is of note that Cu-ZnO/Al_2_O_3_-imp did not produce methanol in the range from 200 to 300 °C, with CO and CH_4_ being the only products formed (Appendix A). Appendix A present chromatograms of the reaction mixture to illustrate the quality of the analysis.

To determine how far from the equilibrium the catalytic values reached for catalysts **2**–**4**, equilibrium data were calculated by minimization of the Gibbs free energy of the system using the RGIBBS module of the Aspenplus© program. For simplicity, these calculations consider the formation of methanol and CO as the only products, ignoring the small percentage of methane and other hydrocarbons detected in the product mixture in very low proportions (less than 5% according to Appendix A). The composition of the equilibrium mixture was determined in the temperature range between 150 and 300 °C, introducing the operation conditions as the starting composition of the system. The results are presented in Table 2. As it can be seen there, a decrease in the CO_2_ conversion from 250 to 350 °C is predicted thermodynamically due to the opposite enthalpy signs of the methanol synthesis (exoergonic, Equation (1)) and the RWGS (endoergonic, Equation (2)). These calculations indicate that the selectivity of methanol 350 °C should be very low. Comparison with the experimental data for catalyst **4** shows that while at 150 and 200 °C, CO_2_ conversion is very far from the equilibrium values, the mixture composition approaches the expected equilibrium values in the temperature range from 250 to 300 °C, with theoretical conversions in the range of 25% and methanol selectivity between 60 and 12% under the operation conditions. 

This catalytic stability of the Cu-ZnO@(N)C samples contrasts with the performance of Cu-ZnO/Al_2_O_3_, obtained following a two-step impregnation procedure similar to that used for the preparation of Cu-ZnO@(N)C (see experimental section). Cu-ZnO/Al_2_O_3_ can be taken as a benchmark catalyst with which one can compare the performance of Cu-ZnO@(N)C samples. The catalytic activity of Cu-ZnO/Al_2_O_3_ is summarized in Appendix A. As can be seen there, the CO_2_ conversion with the fresh Cu-ZnO/Al_2_O_3_ samples was somewhat lower than that achieved by the Cu-ZnO@(N)C samples, with a maximum 31% methanol selectivity at 300 °C at 12% CO_2_ conversion. Therefore, the performance of the Cu-ZnO@(N)C samples compares well with that of fresh Cu-ZnO/Al_2_O_3_ (Appendix A), with selectivity to methanol decreasing with temperature and CO_2_ conversion similarly in both cases. However, it was observed that Cu-ZnO/Al_2_O_3_ undergoes a notable deactivation with time on stream, becoming severely deactivated in a few tens of hours. After 40 h reaction, the spent Cu-ZnO/Al_2_O_3_ catalyst is black in color, suggesting coke deposition during the process (see Appendix A). Combustion chemical analysis of a black deactivated Cu-ZnO/Al_2_O_3_ catalyst showed the presence of over 1% carbon in the material. In the literature, it has been reported that Cu-ZnO/Al_2_O_3_ undergoes deactivation by sintering of the metal NPs [68] and by coke deposition [69]; not surprisingly, this deactivation happens here for the Cu-ZnO/Al_2_O_3_ sample in the time scale of tens of hours. Cu-ZnO/Al_2_O_3_-wp not submitted to pyrolysis behaves similarly, in agreement with the XRD, which shows no difference in the Al_2_O_3_ crystalline phase due to pyrolysis. Thus, activity and stability data confirm a similar performance and much better stability of Cu-ZnO@(N)C with respect to the Cu-ZnO/Al_2_O_3_ reference catalyst.

The above results show the advantage of (N)C as a support in comparison to Al_2_O_3_. To understand the role of (N)C, the type of N atom having a stronger interaction with Cu-ZnO, the CO_2_ adsorption of the Cu-Zn cluster, and the plausible reaction mechanism, DFT calculations were carried out.

### 3.3. Modelling and DFT Calculations

It is well known that the strong metal–support interaction between transition metal clusters and defective carbon (such as N-doped graphene) greatly contributes to the enhancement of the catalytic performance by tuning the electronic structures and improving the stability of, in particular, Cu and Zn-doped Cu clusters [70]. The use of Cu-based catalysts for methanol synthesis via the CO_2_ dehydrogenation route is as old as the process itself and Cu-ZnO/Al_2_O_3_ catalysts are applicable on an industrial scale in the syngas route [66]. On the other hand, CO_2_ hydrogenation to methanol over Cu and Zn-doped Cu clusters supported on graphitic carbon has rarely been reported and its active site is still unclear [71,72,73,74].

To better understand the mechanism of methanol synthesis from CO_2_, periodic DFT calculations on ZnCu/N-C model catalysts were performed. A 5 × 5 super-cell of graphene with one pyridinic-N and one graphitic-N was built (Figure 4), with the two N-doped atoms being far enough away from each other to make their cross interaction negligible. A Cu cluster with 13 atoms (Cu_13_) was simulated (Figure 4a), starting from the experimental geometry of these Cu clusters [75].

The binding strength of Cu_13_ on different sites of N-doped graphene was first examined by periodic DFT calculations. The binding energies of Cu_13_ on the pyridinic-N site, graphitic-N site, and graphitic-C site were calculated to be −5.70, −2.62, and −3.03 eV, respectively (Figure 4c–e). The results show that pyridinic N is the most favorable site for anchoring the Cu_13_ cluster (Appendix A) and the location of Cu_13_ on this site will be used here onwards. This proposal agrees with the Hirshfeld charge distribution (Appendix A), showing a charge transfer from Cu to N-doped graphene of −0.75 e^−^, demonstrating that there is a strong interaction between the Cu cluster and N-doped graphene. Apparently, the lone electron pair of pyridinic N interacts strongly with the Cu_13_ cluster in comparison with graphitic N or C atoms. This strong catalyst–support interaction is believed to provide stability and support to the clusters, preventing their aggregation and maintaining their catalytic activity over multiple reaction cycles.

Additionally to the Cu_13_ cluster, a Zn-doped Cu cluster (Zn_1_Cu_12_) was also considered. The different substituted sites of the Zn_1_Cu_12_ cluster on pyridinic-N of N-doped graphene were geometry optimized (Appendix A). Site 12 exhibits the minimum energy, indicating that it is the most stable structure. Thus, we used this model for subsequent calculations on the mechanism of CO_2_ hydrogenation to methanol. In this model, the Zn atom is at the core surrounded by the twelve external Cu atoms.

To determine the active site of the reaction, different CO_2_ adsorption configurations and sites were computed for the Zn_1_Cu_12_ cluster on the pyridinic-N of N-doped graphene. Appendix A shows that the line-to-line parallel adsorption has the strongest binding energy. Various non-equivalent line-to-line parallel adsorption sites were subsequently considered for CO_2_ adsorption (Appendix A). The results show that one of the Cu-Cu sites (denoted as ‘4-9′, see Appendix A) is the most active for CO_2_ adsorption, compared not only among all Cu-Cu sites, but also among N-C, C-C, or Zn-Cu sites (Appendix A). In this absorption mode, two Cu atoms interact simultaneously with the C and the two O atoms of CO_2_ that are quasi perpendicularly aligned with the graphene sheet. 

Therefore, subsequent intermediates and transition states of CO_2_ conversion to methanol were studied on 4-9 Cu-Cu sites for the Zn_1_Cu_12_ cluster on N-doped graphene. Based on reported studies [76,77], two main reaction pathways for CO_2_ conversion to methanol were considered in our calculations: the (i) RWGS+CO+hydro pathway: RWGS reaction to produce a CO intermediate followed by its hydrogenation to methanol and (ii) formate pathway: initial hydrogenation of CO_2_ to a *HCOO intermediate followed by its hydrogenation and dissociation to methanol. As will be commented below, and in agreement with previous studies [76,77], the present calculations also show the preferred formate pathway on ZnCu/C-N via *HCOOH, *H_2_COOH, and *CH_3_O intermediates over the RWGS+CO-hydro pathway for methanol synthesis (Figure 5 and Appendix A). 

#### 3.3.1. Formate Pathway

HCOO* has been considered as the main intermediate for hydrogenating CO_2_ to CH_3_OH through the formate pathway [65]. Consequently, the rate-determining step is believed to occur in the hydrogenation of HCOO*. Accordingly, our results reveal a low-energy barrier for the formation of HCOO* (0.35 eV, TS1, Appendix A). Then, the rate-determining step is the hydrogenation of *HCOO to form *HCOOH, with an activation energy of 2.12 eV (TS2, Appendix A). Subsequent hydrogenation leads to the formation of *H_2_COOH, with an energy barrier of 0.79 eV (TS3, Appendix A). This is followed by the dissociation of *H_2_COOH into *H_2_CO + *OH, with a barrier of 1.10 eV (TS4, Appendix A), and the hydrogenation of *OH to water, with a barrier of 1.77 eV (TS5, Appendix A). Finally, *H_2_CO is hydrogenated twice to form *H_3_CO (0.84 eV, TS6, Appendix A) and *CH_3_OH (1.56 eV, TS7, Appendix A). These results indicate that the reaction of *HCOO + *H→*HCOOH, with the largest energy barrier (2.12 eV), is the rate-determining step for the synthesis of methanol. Both the low barrier for HCOO* formation and the subsequent largest reaction barrier are lower than the maximum barrier in the RWGS+CO-hydro pathway discussed below, indicating that the formation of methanol should be preferred through the formate pathway.

#### 3.3.2. RWGS+CO+Hydro Pathway

In the RWGS+CO+hydro pathway mechanism, the primary intermediate is CO* instead of HCOO*. Firstly, *CO_2_ is hydrogenated to form *HOCO (1.89 eV, TS1, Appendix A). This is followed by the dissociation of *HOCO (TS2, Appendix A) and the hydrogenation of *OH (TS3, Appendix A), with energy barriers of 2.01 and 1.87 eV, respectively. Then, *CO is hydrogenated four times to form *HCO (0.78 eV, TS4, Appendix A), *H_2_CO (1.91 eV, TS5, Appendix A), *H_3_CO (3.11 eV, TS6, Appendix A), and *CH_3_OH (1.92 eV, TS7, Appendix A). CO is expected to be the main product along the RWGS+CO+hydro pathway, since there is a large energy barrier (3.11 eV, Appendix A) of *H_2_CO hydrogenation to *H_3_CO (TS6). Since the calculated binding energy of *CO is −1.80 eV, desorption is in competition with subsequent reactions such as TS5 with a barrier of 1.91 eV. This will negatively affect the production of methanol through this pathway.

## 4. Conclusions

Although Zn metal undergoes evaporation under pyrolysis conditions, it has been possible to prepare a series of Cu-ZnO@(N)G catalysts in two steps in which, first, chitosan embedding Cu(OAc)_2_ is pyrolyzed and, subsequently, Zn(OAc)_2_ is impregnated and calcined. Since chitosan comes from biomass wastes, the present synthesis represents a clear example of waste valorization and circular economy. The samples show very small metal NPs, about 1 nm average size, well dispersed on the graphitic matrix. These samples act as catalysts for the partial CO_2_ hydrogenation to methanol. The formation of significant proportions of CO, accompanied by lesser amounts of methane and higher hydrocarbons, was also observed. Methanol selectivity decreased with CO_2_ conversion and reaction temperature in the range of temperatures between 150 and 300 °C, as well as depending on the Cu-ZnO@(N)G catalyst. The presence of ZnO in the catalyst was a prerequisite for methanol formation in (N)C support, but the Cu/Zn atomic ratio influenced CO_2_ conversion, rather than methanol selectivity. For the optimal sample, a maximum methanol selectivity of about 89.7% for 1.7% CO_2_ conversion and a 25% selectivity at 21% CO_2_ conversion and a methanol productivity of 83 g_CH3OH_ kg_catalyst_^−1^ h^−1^ were reached. These methanol selectivity values are among the highest reported in the literature. DFT calculations indicate that the presence of pyridinic-N atoms on graphene introduces additional active sites and facilitates the adsorption and binding of CuZn clusters, preventing their aggregation and maintaining their catalytic activity over multiple reaction cycles. According to the more favorable formate pathway, the rate-determining step shows an energy barrier of 2.12 eV. This study opens new possibilities for designing and developing efficient catalysts for CO_2_ conversion to methanol using N-doped graphene as a substrate.

## Data Availability

Data are available from the corresponding authors.

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
