# Peer review of "Nanometric Cu-ZnO Particles Supported on N-Doped Graphitic Carbon as Catalysts for the Selective CO2 Hydrogenation to Methanol"

_nanomaterials, 2024, doi:10.3390/nano14050476_

Round 1

Reviewer 1 Report

Comments and Suggestions for Authors

29.01.2024

The revision of manuscript Nanomaterials entitled:

“Nanometric Cu-ZnO Particles Supported on N-Doped Graphitic Carbon as Catalysts for Selective CO2 Hydrogenation to Methanol”

The manuscript deals with the preparation of Cu-ZnO system deposited on chitosan derived N-doped graphitic carbon support and investigation of Cu/Zn ratio on the fundamental physicochemical properties and catalytic activity toward CO2 conversion to methanol with evaluation of selectivity and long-time on stream stability. Cu modified N-doped carbon (Cu@(N)C) was prepared via a pyrolysis of Cu(OAc)2 doped chitosan hydrogel. Cu-ZnO systems were prepared by Zn(OAc)2 deposition on (Cu@(N)C) and subsequent thermal treatment. For comparative studies the reference Cu-Zn/Al2O3 catalysts were also synthesized in similar way but Al2O3 was used as a support instead of Cu@(N)C. The properties of samples prepared were investigated with several techniques (ICP-OES, XRD, Raman spectroscopy, XPS, DF-TEM, SEM, TG-DTA, SEM/TEM). Finally, their efficiency (activity, selectivity and stability) toward CO2 hydrogenation to methanol in kinetic regime at relatively high GHSV was evaluated.

Authors found ex-chitosan (N)C as efficient support for Cu-ZnO system and 2 step preparation technique as efficient way to deposite nanoparticles of Cu-ZnO system effective toward methanol synthesis by CO2 hydrogenation. The impact of Cu/Zn ratio on the catalyst activity proved to be inconclusive. Another conclusion is that Cu‑ZnO@N(C) systems act as catalysts for just partial CO2 hydrogenation to methanol since formation of substantial amount of CO and minor amount of hydrocarbons was detected in the effluent streams. Finally, the optimal catalyst was establish and its methanol productivity of 83 gCH3OH·kgcatalyst·h-1 was determined.

This work follows the scope of Nanomaterials journal. I found this paper very interesting since novel specific materials preparation was studied and their efficiency toward CO2 to methanol was investigated in detail. However, some of the observations taken seems to be discussed insufficient. I  found some issues which I thought were not satisfactorily explained. The details given below should be reconsidered and corrected:

1.     Synthesis procedure involved hydrogel formation in NaOH aqueous solution but I haven’t found any results concerning residual Na. Since the effect of alkali on Cu-ZnO systems is well known it is recommended to provide residual Na content in final catalysts and to discuss the effect of Na (if any).

2.     Page 3, line 106-107: The motivation to dry the solids in supercritical CO2 after washing them off in anhydrous ethanol should be given. The misunderstanding is deepened by “extraction” instead of “drying” as shown in Figure 1. So what was extracted or removed during supercritical CO2 treatment?

3.     Graphitic form of ex-chitosan (N)C material was revealed by Raman spectra but “.. 2D defective graphene sheets ...” are essentially invisible in (Fig. 2) HD-TEM images (page 7: line 261). One might expect the presence of diffraction peaks characteristic for graphitic carbons at 2θ of 24-27° but XRD patterns (Fig. S1) is limited to just 30°≤2θ80°. I recommend to increase the range of 2θ to 20°≤2θ80° to confirm graphitic form of the (N)C materials.

4.     Sharp diffraction peaks of relative high intensity are present in the XRD patterns (Fig. S1). The value of the work would be increased if Cu and ZnO average crystallite size would be calculated on the base of XRD data according to well known methods and results discussed in the manuscript.

5.     Since Cu‑ZnO@N(C) catalysts contain metallic Cu in dispersed into nanoparticles one might expect pyrophoric properties of materials. Was the pirophoricity observed? Is the risk of partial oxidation by air of carbonaceous materials excluded? What about these issues?

6.     Table 1: What is the meaning of “Average particle size (nm)” determined by DF‑HRTEM? Does “average particle size” to refers to (N)C, Cu, ZnO or other particles? The exact meaning of this parameter should be explained in the section “Sample Characterization”.

7.     Page 5 - Catalyst testing. Grain size of catalyst powder being the sample for activity tests was not given. What fraction of sample was used? I’m afraid that improper parameter was used to calculate Weisz-Prater number. Using “average particle size” provided in Table 1 as “catalyst particle radius” Rp in Weisz-Prater equation is incorrect and leads to false results. Calculation of NW-P number should be reconsidered and corrected properly.

8.     Active Cu surface area (SCu) is one of the key parameters for methanol synthesis catalysts. Since catalytic activity corelates well with SCu that parameter is widely used to comparative studies and evaluation of different catalysts. It is somewhat disappointing that the authors did not measure this parameter (and did not report such data). The scientific value of the manuscript would be higher if authors provided the SCu data for samples synthetizes and compare with literature data for conventional Cu-Zn-Al catalysts and Cu or (Cu-ZnO) deposited on carbonaceous materials.

9.     Page 6 line 238 - Zn-free sample preparation, “… to which Zn(OAc)2 was added ...” should be modified to “… to which Zn(OAc)2 was not added ...”

10.  Page 12, line 468-469: I’m afraid that an assumption of coke deposition phenomena occurrence just on the basis of sample colour might lead to false conclusions since black colour is also characteristic for dispersed metallic Cu. An assumption of coke deposition of spent sample should be supported by sample analysis results.

11.  Numerous editing errors, e.g. improper dividing of words.

For all the reasons above, I recommend major revision of the manuscript.

Author Response

RESPONSE TO THE REVIEWER’S COMMENTS AND DESCRIPTION OF THE CHANGES MADE IN THE REVISION

Title: Nanometric Cu-ZnO Particles Supported on N-Doped Graphitic Carbon as Catalysts for the Selective CO2 Hydrogenation to Methanol

Ref. nanomaterials-2855322

Reviewer 1

The revision of manuscript Nanomaterials entitled:

“Nanometric Cu-ZnO Particles Supported on N-Doped Graphitic Carbon as Catalysts for Selective CO2 Hydrogenation to Methanol”

The manuscript deals with the preparation of Cu-ZnO system deposited on chitosan derived N-doped graphitic carbon support and investigation of Cu/Zn ratio on the fundamental physicochemical properties and catalytic activity toward CO2 conversion to methanol with evaluation of selectivity and long-time on stream stability. Cu modified N-doped carbon (Cu@(N)C) was prepared via a pyrolysis of Cu(OAc)2 doped chitosan hydrogel. Cu-ZnO systems were prepared by Zn(OAc)2 deposition on (Cu@(N)C) and subsequent thermal treatment. For comparative studies the reference Cu-Zn/Al2O3 catalysts were also synthesized in similar way but Al2O3 was used as a support instead of Cu@(N)C. The properties of samples prepared were investigated with several techniques (ICP-OES, XRD, Raman spectroscopy, XPS, DF-TEM, SEM, TG-DTA, SEM/TEM). Finally, their efficiency (activity, selectivity and stability) toward CO2 hydrogenation to methanol in kinetic regime at relatively high GHSV was evaluated.

Authors found ex-chitosan (N)C as efficient support for Cu-ZnO system and 2 step preparation technique as efficient way to deposite nanoparticles of Cu-ZnO system effective toward methanol synthesis by CO2 hydrogenation. The impact of Cu/Zn ratio on the catalyst activity proved to be inconclusive. Another conclusion is that Cu‑ZnO@N(C) systems act as catalysts for just partial CO2 hydrogenation to methanol since formation of substantial amount of CO and minor amount of hydrocarbons was detected in the effluent streams. Finally, the optimal catalyst was establish and its methanol productivity of 83 gCH3OH·kgcatalyst·h-1 was determined.

This work follows the scope of Nanomaterials journal. I found this paper very interesting since novel specific materials preparation was studied and their efficiency toward CO2 to methanol was investigated in detail. However, some of the observations taken seems to be discussed insufficient. I  found some issues which I thought were not satisfactorily explained. The details given below should be reconsidered and corrected:

Response: We thank this reviewer for his/her positive comments and suggestions to improve our manuscript.

  1. Synthesis procedure involved hydrogel formation in NaOH aqueous solution but I haven’t found any results concerning residual Na. Since the effect of alkali on Cu-ZnO systems is well known it is recommended to provide residual Na content in final catalysts and to discuss the effect of Na (if any).

Response: We thank the reviewer for bringing this important point to our attention. However, due to the water solubility of NaOH and the conditions of the pyrolysis in which Na+ becomes converted to Na metal that sublimes in the pyrolysis conditions under Ar flow, no Na content can be determined in the present catalysts. In other cases (Reference 1) we have observed the presence of Ca and other metals in the samples.

  1. Page 3, line 106-107: The motivation to dry the solids in supercritical CO2after washing them off in anhydrous ethanol should be given. The misunderstanding is deepened by “extraction” instead of “drying” as shown in Figure 1. So what was extracted or removed during supercritical CO2 treatment?

Response: The revised version now clarifies that the supercritical CO2 drying here removes ethanol from the chitosan spheres, while producing a highly porous material with large surface area. The equipment used is normally employed for supercritical CO2 extractions, as for instance in decaffeinated coffee beans. However, in the present case, we are using the system for removing ethanol from chitosan beads. The role of supercritical CO2 to dry chitosan beads is also indicated now in Scheme 3 caption.

  1. Graphitic form of ex-chitosan (N)C material was revealed by Raman spectra but “.. 2D defective graphene sheets ...” are essentially invisible in (Fig. 2) HD-TEM images (page 7: line 261). One might expect the presence of diffraction peaks characteristic for graphitic carbons at 2θ of 24-27° but XRD patterns (Fig. S1) is limited to just 30°≤2θ≤80°. I recommend to increase the range of 2θ to 20°≤2θ≤80° to confirm graphitic form of the (N)C materials.

Response: The reviewer is absolutely right that SEM images (Fig. 2) and dark field TEM images (Fig.1) reveal coarse morphology and metal NP distribution, but they do not show the graphitic nature of the (N)C matrix. Following the review’s suggestion, we provide now a revised version of Fig. S1 in where the diffraction peak showing the graphitic stacking of the defective graphene layers appearing at 2Q about 25o is present.

  1. Sharp diffraction peaks of relative high intensity are present in the XRD patterns (Fig. S1). The value of the work would be increased if Cu and ZnO average crystallite size would be calculated on the base of XRD data according to well known methods and results discussed in the manuscript.

Response: We thank the reviewer for this comment. Using the Scherrer equation we have now presented an estimation of the average particle size of Cu (7.8 nm) and ZnO (25.0 nm).

  1. Since Cu‑ZnO@N(C) catalysts contain metallic Cu in dispersed into nanoparticles one might expect pyrophoric properties of materials. Was the pirophoricity observed? Is the risk of partial oxidation by air of carbonaceous materials excluded? What about these issues?

Response: We have prepared at least 20 of these samples by now, some of them in about 0.5 g of amount. In none of the cases, we have observed pyrophoricity of the samples, a property that, however, was observed in the case of some other metals. A comment on the lack of pyrophoricity has now been added in the experimental section, when describing the preparation of Cu-ZnO@(N)G samples.

  1. Table 1: What is the meaning of “Average particle size (nm)” determined by DF‑HRTEM? Does “average particle size” to refers to (N)C, Cu, ZnO or other particles? The exact meaning of this parameter should be explained in the section “Sample Characterization”.

Response: Table 1 and sample characterization now explain that the average particle size refers to the metal NPs. Due to the similarity of atomic weights, it is not possible to differential between Cu and Zn NPs in TEM.

  1. Page 5 - Catalyst testing. Grain size of catalyst powder being the sample for activity tests was not given. What fraction of sample was used? I’m afraid that improper parameter was used to calculate Weisz-Prater number. Using “average particle size” provided in Table 1 as “catalyst particle radius” Rpin Weisz-Prater equation is incorrect and leads to false results. Calculation of NW-P number should be reconsidered and corrected properly.

Response: The referee is correct. Supporting information has now given the average particle size of the catalyst, not of the metal NP. This error derives from a previous version of Table 1 that does not correspond to one of this submission.

  1. Active Cu surface area (SCu) is one of the key parameters for methanol synthesis catalysts. Since catalytic activity corelates well with SCuthat parameter is widely used to comparative studies and evaluation of different catalysts. It is somewhat disappointing that the authors did not measure this parameter (and did not report such data). The scientific value of the manuscript would be higher if authors provided the SCu data for samples synthetizes and compare with literature data for conventional Cu-Zn-Al catalysts and Cu or (Cu-ZnO) deposited on carbonaceous materials.

Response: We agree with this reviewer that Cu dispersion would give an important information on the number of atoms exposed to the reagents. However, these measurements with our equipment require a large amount of sample, typically about 500 mg to be meaningful. This amount of sample was not routinely prepared in our pyrolytic system. In addition, pelletization is also very difficult with the graphene carbons of the study due to static electricity charging, thus limiting the possibility to determine Cu dispersion measurements. We will consider this possibility in our future works.

  1. Page 6 line 238 - Zn-free sample preparation, “… to which Zn(OAc)2was added ...” should be modified to “… to which Zn(OAc)2 was not added ...”

Response: The error has now been corrected. We thank the reviewer for bringing to our attention.

  1. Page 12, line 468-469: I’m afraid that an assumption of coke deposition phenomena occurrence just on the basis of sample colour might lead to false conclusions since black colour is also characteristic for dispersed metallic Cu. An assumption of coke deposition of spent sample should be supported by sample analysis results.

Response: Combustion chemical analysis of the deactivated Cu-ZnO/Al2O3 catalyst showed the presence of over 1% carbon in the material. These new data are now added in the revision.

  1. Numerous editing errors, e.g. improper dividing of words.

Response: The errors have now been corrected.

For all the reasons above, I recommend major revision of the manuscript.

Reviewer 2 Report

Comments and Suggestions for Authors

Dear Authors, I'm sorry, but no, it's not a good work, the most important thing here is probably "Selective hydrogenation of CO2 to methanol", but high selectivity applies only to the lowest conversions (I'm afraid it's below error)

focusing on material characteristics, the conclusion is: "where is the newness?"

A separate question is why the authors think they are working with nanomaterials? to obtain TEM images you have to destroy the starting material, right? The authors can of course argue that they show a distribution, but such small particles should be analyzed by XRD as signals with a very wide range and low intensity, right?

Author Response

RESPONSE TO THE REVIEWER’S COMMENTS AND DESCRIPTION OF THE CHANGES MADE IN THE REVISION

Title: Nanometric Cu-ZnO Particles Supported on N-Doped Graphitic Carbon as Catalysts for the Selective CO2 Hydrogenation to Methanol

Ref. nanomaterials-2855322

Reviewer 2

Dear Authors, I'm sorry, but no, it's not a good work, the most important thing here is probably "Selective hydrogenation of CO2 to methanol", but high selectivity applies only to the lowest conversions (I'm afraid it's below error)

focusing on material characteristics, the conclusion is: "where is the newness?"

A separate question is why the authors think they are working with nanomaterials? to obtain TEM images you have to destroy the starting material, right? The authors can of course argue that they show a distribution, but such small particles should be analyzed by XRD as signals with a very wide range and low intensity, right?

Response: As indicated in the Introduction while commenting the state of the art, selective CO2 hydrogenation to methanol is an important catalytic process that still requires appropriate catalysts. The present manuscript reports that N-doped graphitic carbon is a suitable support for Cu-ZnO particles exhibiting a methanol selectivity among the highest ever reported.

The novelty of the present study is the use of an N-doped carbon derived from biomass waste as support, obtaining a catalyst that is stable over 60 h time on stream.

The dimensions of the Cu/ZnO particles corresponding to the active sites in the range of 1 nm make us to think that we are working with nanomaterials. TEM images are obtained by depositing a micro drop suspension of catalyst on the holder without altering metal NPs.

XRD has now been used to determine average particle size of metal crystals. This average by XRD is typically larger than the one determined by TEM, since in XRD larger particles give stronger diffraction. TEM is considered a more accurate and powerful techniques.

Reviewer 3 Report

Comments and Suggestions for Authors

Manuscript entitled Nanometric Cu-ZnO Particles Supported on N-Doped Graphitic Carbon as Catalysts for the Selective CO2 Hydrogenation to  Methanol has been submitted by Lu Peng and co-authors. Presented research are well designed and complete. All results are logic and consistent. Paper will be interesting for a readers. My detailed comments are given below.

1) In Introduction: Author should make a recent literature review on g-C3N4 materials for catalytic production of methanol

2) In Introduction: Authors should describe also photocatalytic approach - other promising technology for alternative green fuels production (methanol, hydrogen) using Doped Graphitic Carbon modified by Cu or Zn, for example based on the following references and other recent literature. 

- Wojtyła et al, Journal of Inorganic and Organometallic Polymers and Materials 28, Pages 492 - 499, DOI 10.1007/s10904-017-0733-3

- Xu, Xuan et al, Journal of Colloid and Interface Science, 651, pp. 669–677, 10.1016/j.jcis.2023.08.033 

3) Synthesis of material included organic solvents. Authors should discusse or demonstrate the issue related with residuals of organic that may affect obtained catalytic results. Moreover, graphitic carbon nitride can be decomposed during catalytic test and material itself can be a potential source of carbon. Bothe these issues should be discussed and clarified.

4) Caption to figure 3 - authors should explain what "the blank test" means

5) GC data should be provided in manuscript for at least one catalytic test as well as blank experiment.

Comments on the Quality of English Language

Minor editing of English language required

Author Response

RESPONSE TO THE REVIEWER’S COMMENTS AND DESCRIPTION OF THE CHANGES MADE IN THE REVISION

Title: Nanometric Cu-ZnO Particles Supported on N-Doped Graphitic Carbon as Catalysts for the Selective CO2 Hydrogenation to Methanol

Ref. nanomaterials-2855322

Reviewer 3

Manuscript entitled Nanometric Cu-ZnO Particles Supported on N-Doped Graphitic Carbon as Catalysts for the Selective CO2 Hydrogenation to  Methanol has been submitted by Lu Peng and co-authors. Presented research are well designed and complete. All results are logic and consistent. Paper will be interesting for a readers. My detailed comments are given below.

Response: We are very much thankful to this reviewer for his/her positive comments to accept this manuscript after some revision.

1) In Introduction: Author should make a recent literature review on g-C3N4 materials for catalytic production of methanol

Response: As suggested by this reviewer, the use of g-C3N4 materials for catalytic production of methanol is now included in the revised version with a suitable reference.

2) In Introduction: Authors should describe also photocatalytic approach - other promising technology for alternative green fuels production (methanol, hydrogen) using Doped Graphitic Carbon modified by Cu or Zn, for example based on the following references and other recent literature. 

- WojtyÅ‚a et al, Journal of Inorganic and Organometallic Polymers and Materials 28, Pages 492 - 499, DOI 10.1007/s10904-017-0733-3

- Xu, Xuan et al, Journal of Colloid and Interface Science, 651, pp. 669–677, 10.1016/j.jcis.2023.08.033 

Response: Following the reviewer advise, a mention to the photocatalytic methanol synthesis has been made in the revision and the suggested references added to the reference list.

3) Synthesis of material included organic solvents. Authors should discusse or demonstrate the issue related with residuals of organic that may affect obtained catalytic results. Moreover, graphitic carbon nitride can be decomposed during catalytic test and material itself can be a potential source of carbon. Bothe these issues should be discussed and clarified.

Response: We appreciate the comment. However, since the experiments carried out under continuous flow in which only 40 mg of catalyst is introduced in the system that operates with 4 mL of reagents and the each experiment last at least 5 h ensures that methanol, another products cannot derived from the catalyst. A comment on this has been introduced on the experimental section when describing catalyst testing.

4) Caption to figure 3 - authors should explain what "the blank test" means

Response: Fig. 3 caption now clarifies that the blank test means a reaction in the absence of catalyst.

5) GC data should be provided in manuscript for at least one catalytic test as well as blank experiment.

Response: As requested by the reviewer, supporting file now provides a GC trace of the reaction products.

Round 2

Reviewer 1 Report

Comments and Suggestions for Authors

I went through the revised version and particularly the response to the reviewers' comments and remarks. In my opinion sufficient corrections were done. Moreover sufficient statements and explanations were provided causing the discussion of results consistent. 

Author Response

We thank this reviewer for his/her positive recommendations to accept our work.

Reviewer 2 Report

Comments and Suggestions for Authors

Dear Authors, Dear Editor I'm forced to support my previous decision: the material is not selective, novelty is poor, and characterization needs deeper discussion, 

Author Response

Dear Editor,

We thank for your time and efforts in handling our manuscript that was submitted for publication to Nanomaterials. The original version was examined by three experts in the field and based on their comment and your own evaluation, you recommended major revision. In the revision, we considered all the reviewer’s comments and revised the manuscript to address the reviewer’s concerns. The revised version went through a second round by the three reviewers. While reviewers 1 and 3 were satisfied with the changes made and recommend publication of the revised version, reviewer 2 still maintains her/his initial opinion recommending major changes due to the following issues. “selective, novelty is poor, and characterization needs deeper discussion”. We notice, however, that in her/his first report the reviewer commented also that our catalyst containing nanoparticles was not a nanomaterial, that we break the material while taking the TEM images and she/he wanted XRD patterns. It was non-sense that placing the material on the TEM sample holder can break a catalyst into nanoparticles. It also reveals a considerable lack of knowledge to not consider a catalyst having nanoparticles as a nanomaterial, since it is well-known that catalytic properties derive from the small nanometric size of the metal particles, exposing a large percentage of surface metal atoms. Regarding XRD to determine the average particle size, we included this data in the revision.   

Now in the second round, reviewer 2 still insists in the … “selective, novelty is poor, and characterization needs deeper discussion”. Regarding novelty and the significant relevance of our work respect to related materials reported in the literature it should be commented that:

  1. Supported Cu-Zn/reduced graphene oxide (rGO) catalysts have been prepared and its catalytic performance is reported for a hydrogenation of CO2 to methanol. The observed catalytic data indicate that rGO nanosheets are highly beneficial to disperse bimetallic Cu-Zn particles and the employment of 10 wt%CuZn/rGO catalyst afforded 424 mgMeOHgcat−1 h−1 at 250 °C (Deerattrakul, P. Dittanet, M. Sawangphruk, P. Kongkachuichay, J. CO2 Util. 2016, 16, 104-113).
  2. One novelty of our work is to avoid the use of rGO, preparing Cu nanoparticles and the carbon support by pyrolysis in a single step starting from biomass waste. This represents a clear case of waste valorisation.
  3. Later, the same group further investigated the influence of nitrogen doped into carbon nanotubes and graphene in the CO2hydrogenation to methanol to understand the role of the nitrogen species. The achieved catalytic data indicate that 15%CuZn loaded on nitrogen doped graphene aerogel afforded 405.49 mg gcat−1 h−1 methanol yield. Interestingly, this work has reported that the increase in the pyridinic nitrogen content is highly beneficial to promote the methanol production (V. Deerattrakul, N. Yigit, G. Rupprechter, P. Kongkachuichay, Appl. Catal. A: Gen. 2019, 580, 46-52).
  4. It is worth noting that our work has also reached similar conclusions in terms of active sites. “DFT calculations indicate that the presence of pyridinic-N atoms on graphene introduces additional active sites facilitates the adsorption and binding of CuZn clusters, preventing their aggregation and maintaining their catalytic activity over multiple reaction cycles.”
  5. In addition, our work provides in-depth details in the characterization of the as-prepared materials as well as a thorough theoretical calculations to identify the real active sites to support the observed catalytic performance.

We strongly believe that this work would be highly valued among the scientific communities to understand the role of carbon materials in stabilizing the active sites and to improve methanol selectivity by related type of materials. To further explain the novelty and reinforce the methanol selectivity of our material, Abstract, Introduction and Conclusions have been now modified to reinforce these important points. The changes made in these second revision have been highlighted in the manuscript.

Looking forward to hearing from you.

With regards

Hermenegildo Garcia

Reviewer 3 Report

Comments and Suggestions for Authors

Revised manuscript is ready to publication. All the most important changes has been made, according to revision report. Authors have successfully navigated through the revision process, crafting an article that is not only well-researched and informative but also coherent and ready for publication. I recommend its publication without hesitation.

Author Response

(The authors gave the same response as above.)

Round 3

Reviewer 2 Report

Comments and Suggestions for Authors

---

Author Response

This reviewer is somewhat more positive now regarding the TEM analysis and the effort that we did in the work. She/He is still insisting in that we should stress the novelty. We have now added in the abstract and at the end of the introduction, the interest in biomass waste valorization and the notable results on methanol productivity.